# Effects of Growing Cycle and Genotype on the Morphometric Properties and Glucosinolates Amount and Profile of Sprouts, Microgreens and Baby Leaves of Broccoli (*Brassica oleracea* L. var. *italica* Plenck) and Kale (*B. oleracea* L. var. *acephala* DC.)

Maria Concetta Di Bella [1], Stefania Toscano [1,*], Donata Arena [1], Diego A. Moreno [2], Daniela Romano [1] and Ferdinando Branca [1]

[1] Department of Agriculture, Food and Environment (Di3A), Università degli Studi di Catania, Via Valdisavoia 5, 95123 Catania, Italy; maria.dibella@unict.it (M.C.D.B.); arenadonata96@gmail.com (D.A.); dromano@unict.it (D.R.); fbranca@unict.it (F.B.)

[2] CEBAS-CSIC, Phytochemistry and Healthy Foods Lab, Food Science and Technology Department, Campus Universitario de Espinardo-25, E-30100 Murcia, Spain; dmoreno@cebas.csic.es

\* Correspondence: stefania.toscano@unict.it

**Abstract:** Some new foods (sprouts, microgreens and baby leaf) of the brassica genus are appreciated for their nutritional and nutraceutical values. The aim of this experimental trial was to improve the nutraceutical traits of these foods by evaluating the effects of the climatic condition, genotype, and plant growth stage on the development of greater quality in relation to these new foods. The morphometric and glucosinolates (GLSs) traits of three traditional Italian cultivars of *Brassica oleracea* crops, such as broccoli (*B. oleracea* var. *italica*), namely the traditional Sicilian landrace 'Broccolo Nero' (BN) and the commercial 'Cavolo Broccolo Ramoso Calabrese' (CR), as well as the commercial kale cultivar 'Cavolo Laciniato Nero di Toscana' (CL) (*B. oleracea* var. *acephala* DC.), were evaluated in an unheated greenhouse in Catania during the end of 2019 and the beginning of 2020. Plant growth was studied at different phenological stages—from seeds to sprouts, microgreens, and baby leaves—over two growing cycles, one in autumn–winter and the other in spring–summer. 'Broccolo Nero' (BN) broccoli showed more rapid growth and biomass production than the other two cultivars evaluated (i.e., weight, hypocotyl length, and leaf width). The GLS profile varied significantly ($p < 0.05$), in relation both to plant's growth stage and to genotype. The highest amount of glucoraphanin was detected for BN microgreens and baby leaves, about 8 µmol g$^{-1}$ d.w., whereas glucobrassicin and its related derivatives were about 14 µmol g$^{-1}$ d.w. in microgreens and baby leaves of CL and about 15 µmol g$^{-1}$ d.w. and 10 µmol g$^{-1}$ d.w. for glucoraphanin in CR, respectively. These new foods can also be produced at home with simple and cheap equipment

**Keywords:** Brassicaceae; vegetable crops; germplasm exploitation; new foods; greens

## 1. Introduction

Over the last 30 years, consumers have shown a preference for natural foods (rich in bioactive compounds) while looking for a balanced and healthy diet, and they have moved steadily away from the use of artificial chemicals in foods [1]. Consumers choose foods based not only on taste but also on health and wellbeing benefits, and food supply chains support this trend with new competitive and innovative personalized products, such as edible sprouts, microgreens, and baby leaf salads [2]. A current popular strategy is the improvement of dietary habits to manage and prevent non-communicable diseases by innovating with new products, adding value to the agro-biodiversity at a local level,

and thus reducing environmental impacts and providing ready-to-eat options for vegetables [3]. In addition to fresh vegetable consumption, the food industry aims to produce functional new foods for the promotion of health. These products are of great interest due to them being sources of antioxidants, such as glucosinolates, phenolic compounds, vitamins, and minerals [4]. Edible sprouts (germinating seeds a few days old) and microgreens (young vegetables a few weeks old with the presence of true leaves) are healthy and novel fresh foods due to their abundance of nutrients and bioactive compounds, and they may have positive impacts on human health. In particular, the bioactive compounds in cruciferous foods are inducers of cell antioxidant protection against cancer and different chronic diseases [5]. These characteristics can be improved by growing sprouts under optimal temperature conditions [6–8] or by modulating light conditions (LED lights) and temperature regimes [7,9].

Another widely consumed product in the human diet is leafy vegetables; these are important and sought after because they are highly nutritious and are present in various markets with different characteristics (color, taste, and texture). In recent years, several leafy vegetable genotypes have become more popular as fresh cut or baby leaves. Since the demand for ready-to-eat and convenient fresh foods started to increase [10], species belonging to the Brassicaceae family have become globally popular due to ongoing trends related to healthy eating and the consumption of vegetables [11,12]. Generally accepted as rich sources of health-promoting phytochemicals [13,14], Brassicaceae crops includes vegetables that are consumed worldwide, such as broccoli (*Brassica oleracea* L. var. *italica* Plenck) and kale (*Brassica oleracea* L. var. *acephala* DC), both of which are rich in glucosinolates (GLSs), minerals, carotenoids, and vitamins [15,16].

The GLSs and their degradation products (isothiocyanates, ITCs) have been widely studied as important anticancer compounds, but they are also part of plant defense systems against insects and diseases; they are also, at least in part, responsible for the characteristic 'flavor' (pungency, bitter taste, smell, etc.) of these cruciferous vegetables [17–20]. The chemical structure of the GLSs has a common β-D-glucopyranose residue linked through a sulfur atom to A (Z)-N-hydroximinosulfate ester and to a variable side chain (R) derived from one of the following amino acids: alanine, valine, leucine. isoleucine, phenylalanine, methionine, tyrosine, and tryptophan [21].

The production of GLSs, as well as their contents and profiles in several distinguished chemotypes, is influenced by genotypes, environmental growth conditions, cultivation methods, and interactions among these factors [22]. Individual glucosinolate concentrations also change during plant development and in relation to environmental factors, such as temperature regime [23]. The influence of environmental conditions assures interest because innovative products, such as sprouts, microgreens, and baby leaves, can be obtained all year around in different climatic and crop conditions. The content of bioactive compounds also varies in different phenological phases and in terms of the effects of abiotic stresses, such as salinity level [24].

Another element that can modify the GLS profile is genotype; in this framework, analyses of the GLS composition of underexploited landraces and new varieties of *B. oleracea* crops should be conducted through a more exhaustive evaluation of their agronomic performances and biological value for human health in a wider range of environmental conditions, such as the Mediterranean and Atlantic areas [25,26].

In relation to broccoli and kale specifically, we paid attention to three Italian traditional varieties/landraces within the framework of the H2020 BRESOV project (Available online: www.bresov.eu, accessed on 15 June 2021). Two of these were of broccoli typology, 'Broccolo nero', 'Cavolo Broccolo Ramoso Calabrese', and one of them was kale, 'Cavolo Laciniato Nero di Toscana'. Recent studies have shown that *B. oleracea* landraces are often characterized by a higher concentration of nutrients than commercial cultivars. The protection and the enhancement of agro-biodiversity represent a great opportunity for current challenges in terms of food security with high quality and composition standards.

Nevertheless, the adoption of hybrids such as F1 for these crops, due to their rapid growing cycles, high productivity, post-harvest duration, and organoleptic and sensory characteristics, has led to a reduction in their nutritional and phytochemical content [27,28].

'Broccolo Nero' (BN) is a neglected landrace grown in two towns located on the slope of Mt. Etna Sicily [29,30]. This variety is characterized by the absence of plant apical dominance; young shoots provide tender leaves and small inflorescences, and the stem, leaves, and inflorescences are rich in anthocyanins that turn the stems and the leaf midribs and veins a dark violet color, thus making its vegetative and reproductive organs black. 'Cavolo Lacinato Nero di Toscana' (CL), also known as 'Tuscan Kale' or 'Lacinato Kale', is a well-known landrace of kale used in the Italian gastronomic tradition. 'Cavolo Broccolo Ramoso Calabrese' (CR) is a cultivar that produces wide branched inflorescence. All of these cultivars represent rich dietary sources of antioxidants for use in innovations in vegetable production.

In this framework, the objective of this work was to evaluate variations in plant morphometric parameters and the GLS profiles of three studied genotypes, namely BN, CL, and CR, in relation to three plant growth stages and two growing cycles during the autumn-winter and spring-summer seasons. Moreover, we sought to individuate the climatic conditions, genotype, and plant growth stage that were optimal for improving the nutraceuticals traits of these proposed new foods.

## 2. Materials and Methods

### 2.1. Plant Material and Experimental Conditions

We compared two cultivars of broccoli, namely the Sicilian landrace of sprouting broccoli (*Brassica oleracea* L. var. *italica* Plenck) 'Broccolo Nero' (BN) of the Di3A active collection of Catania University and the standard commercial cultivar of broccoli 'Cavolo Broccolo Ramoso Calabrese' (CR) provided by the SAIS s.p.a. seed company (S.A.I.S Sementi S.p.a., Cesena (FC), Italy). We also examined one kale cultivar (*B. oleracea* L. var. *acephala*), which was represented by 'Cavolo Lacinato Nero di Toscana' (CL). They are listed in the vegetable germplasm repository of the Department of Agriculture, Food and Environment (Di3A) at the University of Catania (BN = UNICT 4939 BR 354; CR = UNICT 4960 BR 325; CL = UNICT 4961 BH86).

The cultivars were utilized for two established growing cycles, the first of which started during the autumn season (November 2019), while the second began in spring (April 2020). Seeds were sown in cellular trays using organic substrate (Terri® Bio, Agrochimica S.p.A., Bolzano, Italy) placed in an unheated greenhouse in Catania (South Italy, 37°31′10″ N 15°04′18″ E; under natural light, 4.6 to 9.2 MJ m$^{-2}$ d$^{-1}$ during the first growing cycle and the second growing cycle). The mean temperature registered during the first cycle was $15.4 \pm 5.8$ °C, from November to January, and, during the second cycle, the mean temperature was $22.6 \pm 11.4$ °C from April to June. The plants were collected at three different stages of their growth: sprouts were collected at the cotyledon disclosure without coats and roots; microgreens were collected at the appearance of the first leaf; and baby leaves were collected at the 3–4th true leaf growth phase. After harvesting, plant samples were frozen at −80 C° and they were freeze-dried and ground to obtain a fine powder. Then, they were stored in glass jars at −20 C° for glucosinolate analysis.

### 2.2. Morphometric Parameters

The seed lots, previously characterized in [7], were utilized for the two growing cycles, and the plants were characterized for their main morphometric parameters (weight of 10 individuals, hypocotyl length, and cotyledon length, as well as width of the sprouts). In addition, the length and width of the microgreens, as well as the number, length and width of the true leaves, were determined for the baby leaves.

### 2.3. Glucosinolate Analyses

2.3.1. HPLC-DAD Analyses

Intact glucosinolates analyses were carried out at the CEBAS CSIC laboratory, in the Department of Food and Science Technology (Murcia, Spain). The freeze-dried samples of BN, CL, and CR were used for the extraction of glucosinolates. For each freeze-dried sample, we utilized 100 mg of powder, which was dissolved in 1.5 mL 70% MeOH (Sigma–Aldrich Chemie GmbH, Steinheim, Germany) at 70 °C for 30 min, mixed by vortex every 5 min to facilitate the extraction, and then centrifuged (10,000 rpm, 15 min, 4 °C). Supernatant was collected, and the methanol was completely removed using a rotary evaporator under vacuum at 37 °C. The dry material obtained was dissolved again in 1 mL of ultrapure water and filtered through a 0.22 μm pore size PVDF syringe filters.

2.3.2. HPLC-DAD-ESI-MSn Analysis of GLSs

For identification and quantification of glucosinolates, we used a HPLC-PDA-ESI-MSn system (Agilent 1200 HPLC-DAD, Barcelona, Spain) coupled with a Bruker MS Ion Trap (Bremen, Germany). Chromatographic conditions and identification and quantification techniques are available elsewhere [31].

### 2.4. Statistical Analysis

The experimental split plot design considered 3 main plots (genotypes) and 2 sub plots (growing cycle) with 3 replications. A multifactorial ANOVA was performed to evaluate the effects of genotype and growing cycle on morphometric plant characteristics. The mean values associated with the main factors, as well as their interactions, were evaluated using Tukey's test ($p < 0.05$). The significance of differences between glucosinolate compounds for each genotype was evaluated by one-way analysis of variance (ANOVA). Data were reported as mean ± standard error (S.E.). All statistical analyses were performed using CoStat release 6.311 (CoHort Software, Monterey, CA, USA).

## 3. Results

### 3.1. Plant Characteristics

The morphometric characteristics of the sprouts are showed in Table 1. The weight of the sprouts did not show any significant difference with regard to either genotype or growing cycle. Hypocotyl length was significantly affected by genotype, as well as by the interaction between growing cycle and genotype. The highest value was observed for CR during the spring growing season (45.32 ± 0.12 mm), while the lowest value was observed for CL during both growing cycles (29.2 ± 3.61 and 31.34 ± 0.40 mm, respectively, for the first and second growing cycle). Cotyledon length was significantly affected by both growing cycle and genotype, as well as by their interaction. The highest values were observed for CR during the spring growing cycle (16.34 ± 0.23 mm), whereas the lowest values were registered for BN and CL during the first growing cycle (10.61 ± 0.24 and 10.30 ± 0.81 mm, respectively). Cotyledon width reached its highest values for CR during the second growing cycle (18.34 ± 0.16 mm), while the lowest value was observed for CL during the first growing cycle (12.2 ± 0.81 mm).

**Table 1.** Morphometric characteristics of the sprouts of 'Broccolo Nero' (BN), 'Cavolo Lacinato Nero di Toscana' (CL), and 'Cavolo Broccolo Ramoso Calabrese' (CR), according to the "cycle" and "genotype" factors, as well as their interactions. Data are reported as mean ± S.E. (*n* = 3). W = weight of 10 individuals (g); HL = hypocotyl length (mm); S = cotyledon length (mm); CW = cotyledon width (mm).

| | | W (g) | HL (mm) | S (mm) | CW (mm) |
|---|---|---|---|---|---|
| Cycle (C) | | | | | |
| | I | 1.20 ± 0.11 | 35.97 ± 2.25 b | 11.71 ± 0.76 b | 14.37 ± 0.73 b |
| | II | 1.38 ± 0.09 | 38.79 ± 2.03 a | 13.89 ± 0.64 a | 16.67 ± 0.51 a |
| | | *n.s.* | * | *** | *** |
| Genotype (G) | | | | | |
| | BN | 1.12 ± 0.12 | 40.91 ± 0.64 a | 11.35 ± 0.36 b | 14.81 ± 0.18 b |
| | CL | 1.28 ± 0.10 | 30.27 ± 1.69 b | 11.77 ± 0.75 b | 14.37 ± 1.09 b |
| | CR | 1.47 ± 0.13 | 40.96 ± 2.20 a | 15.28 ± 0.72 a | 17.37 ± 0.66 a |
| | | *n.s.* | *** | *** | ** |
| C × G | | | | | |
| I | BN | 1.03 ± 0.11 | 42.1 ± 0.75 ab | 10.61 ± 0.24 de | 14.5 ± 0.23 bc |
| | CL | 1.16 ± 0.17 | 29.2 ± 3.61 d | 10.30 ± 0.81 e | 12.2 ± 0.81 c |
| | CR | 1.41 ± 0.28 | 36.6 ± 2.31b cd | 14.22 ± 1.22 bc | 16.40 ± 1.10 ab |
| II | BN | 1.22 ± 0.23 | 39.71 ± 0.17 abc | 12.09 ± 0.17 bc | 15.13 ± 0.11 bc |
| | CL | 1.40 ± 0.12 | 31.34 ± 0.40 cd | 13.24 ± 0.12 bc | 16.53 ± 0.75 ab |
| | CR | 1.52 ± 0.07 | 45.32 ± 0.12 a | 16.34 ± 0.23 a | 18.34 ± 0.16 a |
| | | *n.s.* | * | ** | *** |

The mean values associated with the two factors and their interaction were evaluated according to Tukey's test. Values (mean ± S.E.) within each column followed by the same letter do not significantly differ at *p* < 0.05 according to Tukey's test; *n.s.* not significant; * significant at *p* < 0.05; ** significant at *p* < 0.01; *** significant at *p* < 0.001.

The morphometric characteristics of the microgreens are shown in Table 2. The weight of the microgreens did not show any significant differences in relation to either genotype or experimental growing cycle factors. Hypocotyl length showed a significant interaction between genotype and growing cycle. The highest value was recorded for BN during the second cycle and the lowest values were recorded for CL and CR during the spring cycle. Cotyledon length was only affected by genotype, while BN showed the highest value (33.36 ± 0.49 mm). Leaf length was significantly affected by both experimental factors, as well as by their interaction. The highest value was measured in BN in the second cycle (38.59 ± 1.45 mm). Leaf width increased significantly during the second cycle.

**Table 2.** Morphometric characteristics of the microgreens of 'Broccolo Nero' (BN), 'Cavolo Lacinato Nero di Toscana' (CL), and 'Cavolo Broccolo Ramoso Calabrese' (CR), according to the "cycle" and "genotype" factors, as well as their interaction. Data are reported as mean ± S.E. (*n* = 3). W = weight of 10 individuals (g); HL = hypocotyl length (mm); S = cotyledon length (mm); CW = cotyledon width (mm); LL = leaf length (mm); LW = leaf width (mm).

| | | W (g) | HL (mm) | S (mm) | CW (mm) | LL (mm) | LW (mm) |
|---|---|---|---|---|---|---|---|
| Cycle (C) | | | | | | | |
| | I | 2.99 ± 0.22 | 33.91 ± 2.63 b | 27.19 ± 1.43 | 19.50 ± 1.14 | 25.51 ± 1.25 b | 17.68 ± 0.87 b |
| | II | 3.49 ± 0.22 | 42.97 ± 3.83 a | 28.61 ± 1.54 | 20.84 ± 1.43 | 31.29 ± 2.01 a | 20.75 ± 0.39 a |
| | | *n.s.* | *** | *n.s.* | *n.s.* | ** | ** |
| Genotype (G) | | | | | | | |
| | BN | 3.48 ± 0.22 | 50.31 ± 3.15 a | 33.36 ± 0.49 a | 24.45 ± 0.63 a | 32.10 ± 3.15 a | 18.97 ± 1.32 |
| | CL | 2.81 ± 0.19 | 30.49 ± 1.32 b | 24.29 ± 0.91 b | 17.91 ± 0.99 b | 24.82 ± 1.04 b | 18.50 ± 0.64 |
| | CR | 3.43 ± 0.36 | 34.51 ± 2.94 b | 26.05 ± 0.77 b | 18.16 ± 1.18 b | 28.27 ± 1.44 ab | 20.17 ± 1.08 |
| | | *n.s.* | *** | *** | *** | ** | *n.s.* |
| C × G | | | | | | | |
| I | BN | 3.42 ± 0.46 | 43.30 ± 0.64 b | 32.40 ± 0.40 | 23.10 ± 0.35 | 25.62 ± 2.36 b | 16.43 ± 1.50 |

| | | | | | | | |
|---|---|---|---|---|---|---|---|
| | CL | 2.72 ± 0.41 | 29.54 ± 2.62 d | 23.47 ± 0.20 | 17.60 ± 0.98 | 23.70 ± 1.73 b | 17.58 ± 0.95 |
| | CR | 2.82 ± 0.28 | 28.90 ± 3.18 d | 25.70 ± 1.62 | 17.80 ± 2.18 | 27.20 ± 2.66 b | 19.02 ± 2.07 |
| II | BN | 3.53 ± 0.17 | 57.33 ± 0.23 a | 34.31 ± 0.35 | 25.81 ± 0.17 | 38.59 ± 1.45 a | 21.51 ± 0.17 |
| | CL | 2.90 ± 0.12 | 31.44 ± 0.99 cd | 25.10 ± 1.85 | 18.21 ± 1.96 | 25.94 ± 1.09 b | 19.42 ± 0.52 |
| | CR | 4.03 ± 0.46 | 40.12 ± 1.33 bc | 26.41 ± 0.46 | 18.52 ± 1.45 | 29.34 ± 1.49 b | 21.32 ± 0.46 |
| | | *n.s.* | * | *n.s.* | *n.s.* | * | *n.s.* |

The mean values associated with the two factors and their interaction were evaluated according to Tukey's test. Values (mean ± S.E.) within each column followed by the same letter do not significantly differ at $p < 0.05$ according to Tukey's test; *n.s.* not significant; * significant at $p < 0.05$; ** significant at $p < 0.01$; *** significant at $p < 0.001$.

The morphometric characteristics of the baby leaves are shown in Table 3. The weight of the microgreens varied according to the genotype, and BN showed the highest value (8.97 ± 0.49 g). Stem length was also significantly influenced by genotype fluctuations (123.55 ± 2.47 mm for CR and 128.97 ± 3.46 mm for CL). Leaf number was influenced only by growing cycle (Table 3). The leaf length was affected by genotype, and the highest value was measured for BN and CR (125.02 ± 1.03 and 122.02 ± 0.98 mm, respectively). The leaf width was affected the genotype–growing cycle interaction; the highest value was observed for BN harvested at the end of the growing cycle (48.31 ± 0.87 mm) (Table 3).

**Table 3.** Morphometric characteristics of the baby leaves of 'Broccolo Nero' (BN), 'Cavolo Lacinato Nero di Toscana' (CL), and 'Cavolo Broccolo Ramoso Calabrese' (CR), according to the "cycle" and "genotype" factors, as well as their interaction. Data are reported as mean ± S.E. (*n* = 3). W = weight of 10 individuals (g); SL = stem length (mm); N = number of true leaf (n); LL = leaf length (mm); LW = leaf width (mm).

| | | W (g) | SL (mm) | N (n) | LL (mm) | LW (mm) |
|---|---|---|---|---|---|---|
| Cycle (C) | | | | | | |
| | I | 7.63 ± 0.40 | 118.59 ± 3.88 | 3.38 ± 0.14 b | 119.43 ± 1.86 | 36.82 ± 1.58 b |
| | II | 8.24 ± 0.39 | 122.26 ± 3.49 | 4.68 ± 0.10 a | 121.56 ± 1.59 | 39.33 ± 2.34 a |
| | | *n.s.* | *n.s.* | *** | *n.s.* | * |
| Genotype (G) | | | | | | |
| | BN | 8.97 ± 0.49 a | 108.76 ± 2.36 b | 4.01 ± 0.28 | 125.02 ± 1.03 a | 45.42 ± 1.49 a |
| | CL | 7.27 ± 0.40 b | 128.97 ± 3.46 a | 4.21 ± 0.30 | 114.45 ± 1.05 b | 32.92 ± 0.62 c |
| | CR | 7.57 ± 0.2 ab | 123.55 ± 2.47 a | 3.87 ± 0.38 | 122.02 ± 0.98 a | 35.88 ± 0.60 b |
| | | * | ** | *n.s.* | *** | *** |
| C × G | | | | | | |
| I | BN | 8.42 ± 0.64 | 107.77 ± 3.84 | 3.51 ± 0.23 | 124.32 ± 2.08 | 42.53 ± 1.41 b |
| | CL | 7.05 ± 0.83 | 125.53 ± 6.90 | 3.61 ± 0.29 | 113.28 ± 1.77 | 32.61 ± 0.87 c |
| | CR | 7.42 ± 0.52 | 122.47 ± 5.34 | 3.03 ± 0.04 | 120.71 ± 1.50 | 35.31 ± 0.86 c |
| II | BN | 9.52 ± 0.69 | 109.74 ± 3.47 | 4.50 ± 0.29 | 125.72 ± 0.68 | 48.31 ± 0.87 a |
| | CL | 7.50 ± 0.23 | 132.41 ± 0.75 | 4.82 ± 0.06 | 115.63 ± 1.04 | 33.23 ± 1.04 c |
| | CR | 7.71 ± 0.16 | 124.63 ± 0.87 | 4.71 ± 0.11 | 123.33 ± 0.92 | 36.44 ± 0.87 c |
| | | *n.s.* | *n.s.* | *n.s.* | *n.s.* | * |

The mean values associated with the two factors and their interaction were evaluated according to Tukey's test. Values (mean ± S.E.) within each column followed by the same letter do not significantly differ at $p < 0.05$ according to Tukey's test; *n.s.* not significant; * significant at $p < 0.05$; ** significant at $p < 0.01$; *** significant at $p < 0.001$.

*3.2. Glucosinolate (GLS) Profile*

The analysis of the GLS profiles of the three different genotypes was carried out during the two growing cycles, accounting for seeds, sprouts, microgreens, and baby leaves. The GLSs identified were as follows: aliphatic glucoraphanin (GRA), glucoiberin (GIB), and glucoerucin (GER), the indoles 4-hydroxy glucobrassicin (HGB), glucobrassicin (GBS), 4-methoxyglucobrassicin (MGB), and neoglucobrassicin (NGB). The total content of GLSs is presented as the addition of the individual identified GLSs.

In relation to seeds, total GLSs showed the highest value in CR and CL, whereas the lowest value was observed in BN (Figure 1). MGB and NGB were not detected in the seeds. The major GLS present in the different studied varieties was GRA, while GIB and GBS showed the lowest values. The highest value of total GLSs was found in the sprouts of the second growing cycle of BN, whereas the lowest value was quantified in the microgreens of the first growing cycle (Figure 2). For the sprouts, in both growing cycles, GRA ($6.30 \pm 0.07$ and $7.40 \pm 0.03$ $\mu$mol g$^{-1}$ d.w., respectively, for the first and the second cycle) was the predominant compound. GIB, MGB, and NGB were not detected in the sprouts of the first cycle.

For microgreens, the most representative GLSs were the indolic GBS and NGB in both growing cycles (3.00 and 2.80, and 4.80 and 3.90 $\mu$mol g$^{-1}$ d.w. for the first and the second growing cycle, respectively). For baby leaves, as in the microgreens, GBS and NGB (6.00 and 5.00, and 5.01 and 7.00 $\mu$mol g$^{-1}$ d.w. for the first and the second growing cycle, respectively) showed the highest amount among the detected GLSs. For sprouts and microgreens, GIB was not detected in either growing cycle. MGB was not detected in sprouts or microgreens in the first growing cycle, and nor was it observed in baby leaves during the second growing cycle. Finally, NGB was not detected in microgreens during the second growing cycle (Figure 2).

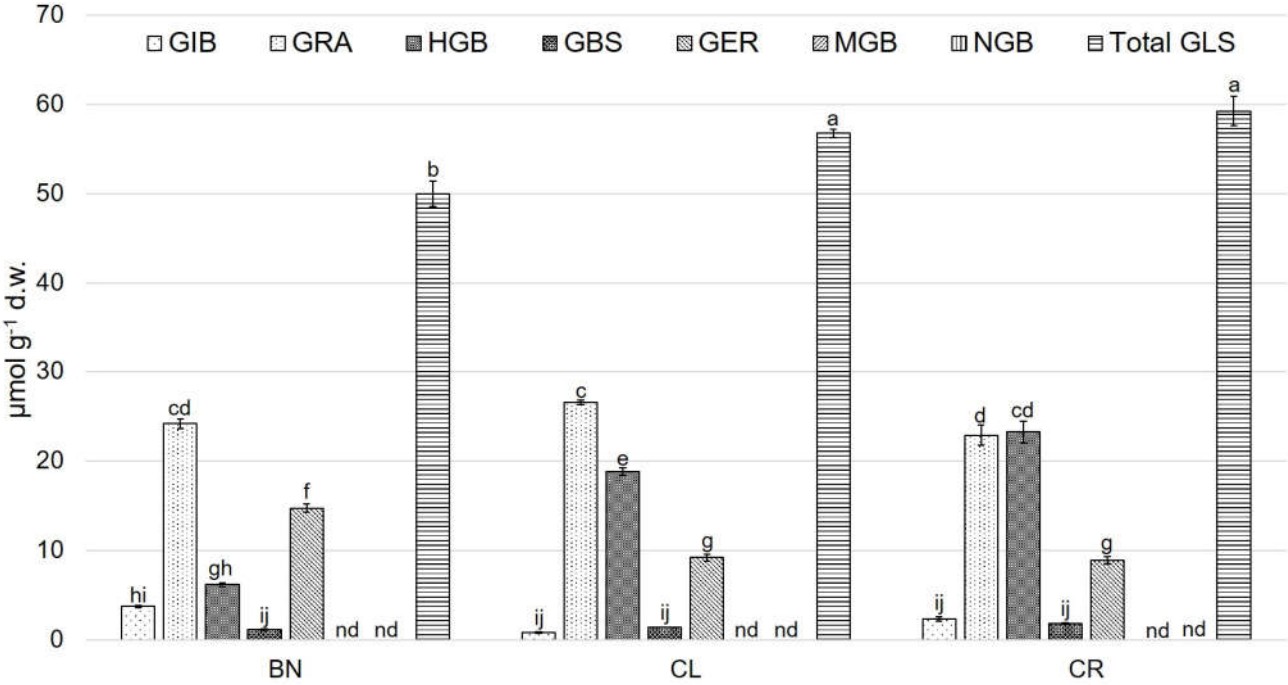

**Figure 1.** Glucosinolate compounds ($\mu$mol g$^{-1}$ d.w.) in seeds of of 'Broccolo Nero' (BN), 'Cavolo Lacinato Nero di Toscana' (CL), and 'Cavolo Broccolo Ramoso Calabrese' (CR). Data are reported as mean ± S.E. Bars with the same letters are not significantly different, as determined by Tukey's test ($p < 0.05$); nd = not detected compound. GIB, glucoiberin; GRA, glucoraphanin; HGB, 4-hydroxyglucobrassicin; GBS, glucobrassicin; GER, glucoerucin; MGB, 4-methoxyglucobrassicin; NGB, neoglucobrassicin; total GLS, total glucosinolates.

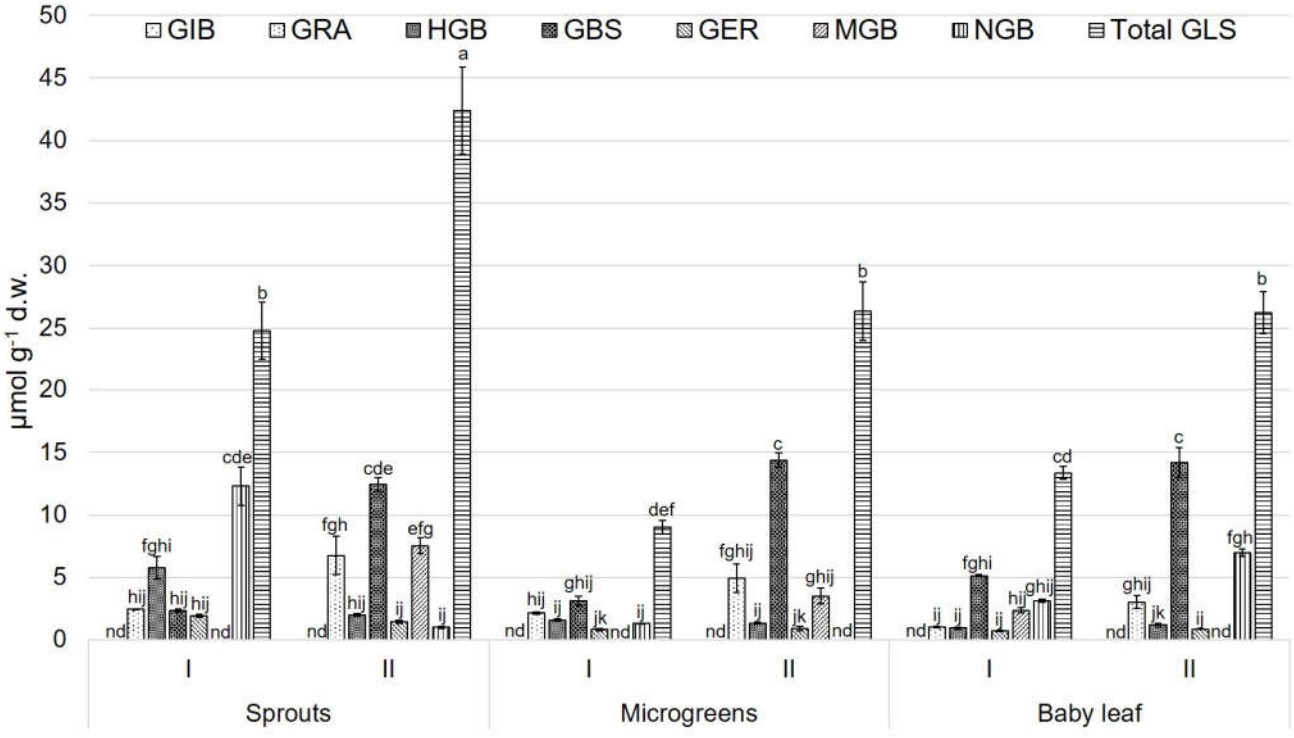

**Figure 2.** Glucosinolate compounds (μmol g$^{-1}$ d.w.) of 'Broccolo Nero' (BN) at different growth stages and during two growing cycles. Data are reported as mean ± S.E. Bars with the same letters are not significantly different, as determined by Tukey's test ($p < 0.05$); nd = not detected compound. GIB, glucoiberin; GRA, glucoraphanin; HGB, 4-hydroxyglucobrassicin; GBS, glucobrassicin; GER, glucoerucin; MGB, 4-methoxyglucobrassicin; NGB, neoglucobrassicin; total GLS, total glucosinolates.

For CL, the highest total GLSs value was observed for the microgreens obtained in the second growing cycle, whereas the lowest content was obtained in baby leaves harvested during the winter cycle (Figure 3). For sprouts, GBS (5.54 ± 0.48 μmol g$^{-1}$ d.w.) dominated during the spring cycle.

The highest amounts of GBS (3.33 ± 0.31 14.76 ± 0.15 μmol g$^{-1}$ d.w., respectively) were shown in microgreens in both growing cycles. For baby leaves, GBS showed the highest value (13.75 ± 0.69 μmol g$^{-1}$ d.w.) during the spring cycle (Figure 3).

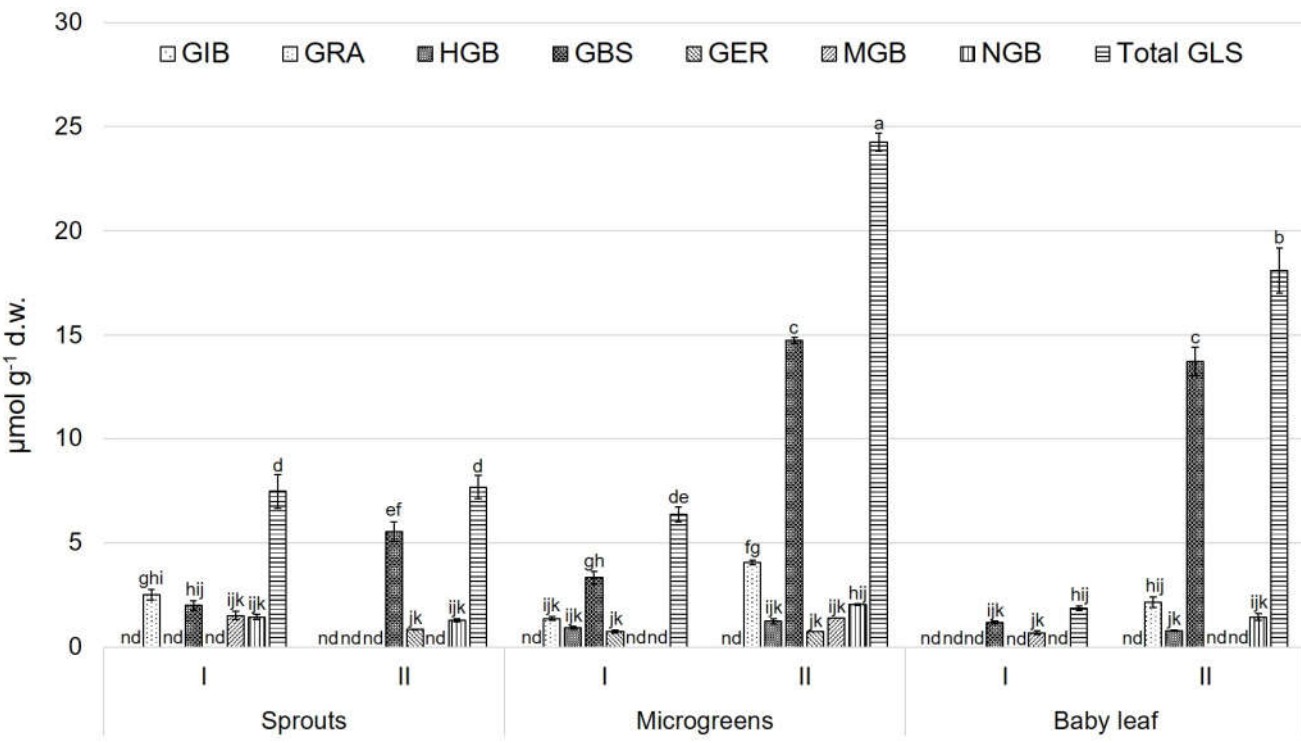

**Figure 3.** Glucosinolate compounds (μmol g⁻¹ d.w.) for 'Cavolo Lacinato Nero di Toscana' (CL) at different growth stages and during two growing cycles. Data are reported as mean ± S.E. Bars with the same letters are not significantly different, as determined by Tukey's test ($p < 0.05$); nd = not detected compound. GIB, glucoiberin; GRA, glucoraphanin; HGB, 4-hydroxyglucobrassicin; GBS, glucobrassicin; GER, glucoerucin; MGB, 4-methoxyglucobrassicin; NGB, neoglucobrassicin; total GLS, total glucosinolates.

For CR, the highest value of GLSs was observed for the sprouts obtained by the spring growing cycle, whereas the lowest ones were observed for microgreens and baby leaves for the spring growing cycle (Figure 4). In the sprouts harvested during the winter growing cycle, NGB (12.28 ± 1.51 μmol g⁻¹ d.w.) dominated; however, in the second cycle, a higher amount was observed for GBS (12.43 ± 0.54 μmol g⁻¹ d.w.). GIB and MGB were not detected in the sprouts of the first growing cycle.

The microgreens in both growing cycles showed 3.13 ± 0.34 and 14.37 ± 0.57 μmol g⁻¹ d.w. of GLSs, respectively. For the baby leaves collected during the first growing cycle, GBS, MGB, and NGB were detected (5.12 ± 0.10, 2.37 ± 0.19 and 3.13 ± 0.10 μmol g⁻¹ d.w., respectively). During the second cycle, the highest amount was observed for the GBS (14.18 ± 1.20 μmol g⁻¹ d.w.) (Figure 4).

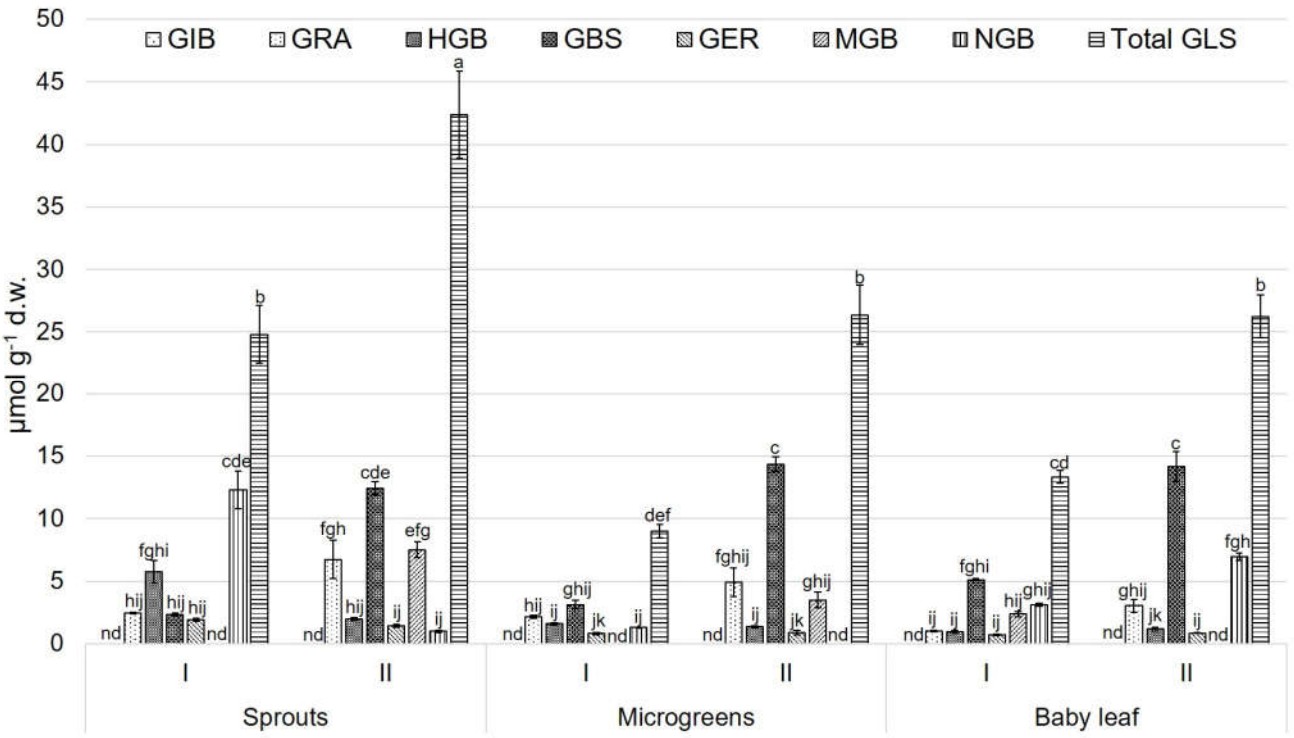

**Figure 4.** Glucosinolate compounds (µmol g⁻¹ d.w.) for 'Cavolo Broccolo Ramoso Calabrese' (CR) at different growth stages and during two growing cycles. Data are reported as mean ± S.E. Bars with the same letters are not significantly different, as determined by Tukey's test ($p < 0.05$); nd = not detected compound. GIB, glucoiberin; GRA, glucoraphanin; HGB, 4-hydroxyglucobrassicin; GBS, glucobrassicin; GER, glucoerucin; MGB,4- methoxyglucobrassicin; NGB, neoglucobrassicin; total GLS, total glucosinolates.

## 4. Discussion

Various typologies of vegetables and herbs, including sprouts, microgreens, and baby leaves, are becoming popular due to their simple production techniques and high nutritional and nutraceutical values [32]. However, increased interest in these new foods is also supported by our results, showing that climatic conditions and plant growth stage could offer the best performance in terms of yield and of GLSs content and profile [6,7]. A considerable variability in the total amount and profile of GLSs was evidenced in this study.

Regarding the plant morphometric characteristics related to the three analyzed growth stages, the results show an increased plant biomass of about 24 times from seeds to sprouts for BN, 32 times for CL and 35-fold for CR, respectively. Comparing sprouts and microgreens, we also observed an increase in plant biomass of three-fold for BN and two-fold for CL and CR. The weight of baby leaves increased during growth by seven- and two-fold for CL and CR, respectively. The baby leaves' biomass increased three-fold for BN and CL and two-fold for CR in comparison to the microgreens, showing similar variation reported by several authors [2,7]. The differences observed among the examined genotypes showed a faster plant growth for BN and CL than for CR. For BN, cotyledon, hypocotyl, and leaf size determined its difference in relation to the other genotypes studied [6,7,24]. Plants during the second growing cycle (spring-summer) showed morphometric characteristics and quantities of total GLSs that were higher than in the first growing cycle (autumn-winter). Several studies have shown that external factors influence GLSs content and composition in Brassicaceae species, thereby enhancing the health-promoting properties of these compounds, which are dependent on quantity and quality [24]. The traditional Sicilian cultivar of BN and the commercial cultivar of CR showed faster

growth in comparison to CL, as demonstrated by the weight of the baby leaves in both genotypes.

Morphometric traits at the different phenological growth phases were associated with the GLSs profiles to favor the availability and consumption of vegetables as sources of health nutrients that can combat cancer and chronic degenerative diseases. The total amount of GLSs was highest (about 55 μmol $g^{-1}$ d.w.) in the seeds of the three cultivars studied, and their values were reduced to less than half in the sprouts, microgreens, and baby leaves, except for the baby leaves of the 'Cavolo Broccolo Ramoso Calabrese' (CR) that was grown during the spring–summer season.

Seed metabolism is activated by germination, which promotes the hydrolysis of carbohydrates through the synthesis of secondary metabolites; this process is of interest in relation to human health [33]. The GLSs amount decreased during the germination process, reaching a high amount until day 4 after cotyledon disclosure, followed by a marked decline between days 4 and 12 (in broccoli, rutabaga, turnip greens, and radish), corresponding to a 50–90% loss of individual GLSs [31,34,35]. In fact, our results highlight a higher content of GLSs in the BN and CR sprouts during the spring–summer cycle. Therefore, sprouts, as stated by many authors, have a greater quantity of antioxidant compounds, mainly glucosinolates [4,36]. According to the literature, our research show that microgreens and baby leaves also have a high content of glucosinolates, but at a lower concentration than that found in sprouts [5,37]. According to several studies, broccoli, cabbage, and kale are good sources of glucosinolates, particularly GRA [7,38]. Therefore, the results of our study are consistent with those previously reported, and they confirm that sprouts are one of the major dietary sources for GRA, which is also present in microgreens and baby leaves of BN; however, GBS is the predominant GLS in CL. The total content of GLSs is driven by the high presence of GRA (4-methylsulfinybutyl-GLS), the precursor of sulforaphane. Moreover, glucoiberin (GIB, 3-methyl-sulfinyl-propyl-GLS) and neoglucobrassicin (NGB, 1-methoxy-3-indolyl-methyl-GLS) were also present in the studied samples; lower amounts of GER, MGB, GBS and HGB were detected; this is a similar trend to that shown in previous studies [39].

The aim of this work was to show that the morphometric traits and GLSs profile were influenced both by the climatic conditions that characterized different plant growth stages and by the genotype taken into consideration [40,41]. In any case, further studies would be needed to better define the relationship between environmental conditions and GLSs profile. Different studies have shown that the GLSs content increases with high temperatures and radiation. According to Shuai-Qi et al. [42], the present research confirms that the GLSs profile is higher in the spring–summer cycle, and therefore increases with high temperatures. Schonhof et al. [43] detected the highest level of major glucosinolates in broccoli by considering the effect of high temperatures; the major compounds found were glucoraphanin and glucobrassicin, confirming what is highlighted in our results. The different GLSs profiles found may be due to the fact that enzymes are influenced by various climatic conditions, such as temperature [44]. The obtained products were influenced by sensory quality and chemical properties, particularly the intensity of sensory attributes such as pungent odor, bitterness and astringency [44]. These characteristics are due to the different concentrations of GLSs; the properties responsible for these concentrations were correlated to the presence of glucoraphanin, which itself had a high concentration in this study.

## 5. Conclusions

Novel foods are becoming more popular for consumers due to their biochemical and organoleptic traits. In accordance with the literature, the present work allowed us to study information and new data related to new foods, such as sprouts, microgreens, and baby leaves, and we confirmed a high amount of antioxidant bioactive compounds, such as glucosinolates, in *B. oleracea* crops. The studied genotypes showed different profiles in

relation to the growth cycle (autumn–winter and spring–summer) and phases of the examined plants. During the spring–summer growing cycle, we obtained the highest yield of the three new foods, as well as the highest amount of GLSs.

The traditional Sicilian cultivar BN and the commercial cultivar CR showed greater biomass production in comparison to CL in both of the studied climatic conditions. The high-quality value of these genotypes could be of interest for organic food chains, especially in traditional rural districts, by establishing the specific denomination of origin and geographic indication labels. On the other hand, as evidenced in this study, and as confirmed by previous works, sprouts showed a greater quantity of GLSs than more developed plantlets, such as microgreens and baby leaves. The potential interest in their production and consumption should be focused on innovative and ecologically sustainable production systems, paying particular attention to their sustainability and socioeconomic impact on organic food chains, as well as on the health of the consumers. In addition, these new products could be grown at home and utilized for fresh consumption and/or home processing (beverage, juices, snacks, etc.).

**Author Contributions:** Conceptualization, F.B. and D.A.M.; methodology, F.B., D.A.M. and D.R.; software, S.T.; validation, F.B., D.A.M. and S.T.; formal analysis, S.T.; investigation, M.C.D.B., S.T. and D.A.; resources, F.B. and M.C.D.B.; data curation, M.C.D.B., F.B., S.T. and D.A.; writing—original draft preparation, M.C.D.B. and S.T.; writing—review and editing, F.B., D.R. and D.A.M.; visualization, S.T. and D.R.; supervision, F.B. and D.R.; project administration, F.B.; funding acquisition, F.B. All authors have read and agreed to the published version of the manuscript.

**Funding:** This research was supported by the project BRESOV (Breeding for Resilient, Efficient and Sustainable Organic Vegetable production) funded by EU H2020 Programme SFS-07-2017. Grant Agreement n. 774244.

**Institutional Review Board Statement:** Not applicable.

**Informed Consent Statement:** Not applicable.

**Data Availability Statement:** Main data are contained within the article; further data presented in this study are available on request from the corresponding author.

**Acknowledgments:** D.A.M. would like to thank also the Fundación Seneca—Murcia Regional Agency for Science and Technology, that partially supported the participation in this research through the Project Ref. 20855/PI/18.

**Conflicts of Interest:** The authors declare no conflict of interest.

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
