# Peer review of "Effects of Growing Cycle and Genotype on the Morphometric Properties and Glucosinolates Amount and Profile of Sprouts, Microgreens and Baby Leaves of Broccoli (Brassica oleracea L. var. italica Plenck) and Kale (B. oleracea L. var. acephala DC.)"

_agronomy, doi:10.3390/agronomy11091685_

Round 1

Reviewer 1 Report

The manuscript is written well and it is of critical importance to stakeholders involved in this specialized crops of emerging market. However, there are some outstanding issues are noticeable and authors are encouraged to address them in the revised version. 

Agronomy 1328925 Manuscript Review Report:

OVERALL REMARKS: The manuscript is written well and it is of critical importance to stakeholders involved in specialized crops of emerging market. However, there are some outstanding issues are noticeable and authors are encouraged to address them in the revised version. Please see the comments mentioned below for different sections.

Comment 1. Additionally, similar abbreviation (CL) is being used for two different terms like “Cavolo Lacinato Nero di Toscana” and “Cotyledon Length”, please assign two different abbreviations respectively and correct it throught the manuscript.

Comment 2. Also, there is unnecessary use of comma on many place throught the manuscript. Please have your work reviewed from native English speaker to address this minor issues.

ABSTRACT: Abstract is written succinctly and it is about in the recommended word count (196 words); however, there are abbreviations (CL and CR) being used without them spelled in the first place of occurance. The year and location of bio-agro-morphological evaluation needs to be mentioned briefly.

INTRODUCTION: Authors have provided good background of related topics succinctly and cited relevant references. However, there are so many mini paragraphs, I encouraged authors to write one succinct medium length paragraphs that explains one idea rather than several mini paragraphs. This can be done to create three medium size paragraphs rather than having 8-10 mini-paragraphs. Also, please make sure to elaborate on research hypothesis in the last paragraph.

Line 54: “are increasing demanding options” please rewrite this as “are increasing in demand” instead

Lines 61-75: Please merge all three mini paragraphs into one since the context of these paragraphs is similar

Lines 76-85: Please merge both mini paragraphs into one since the context of these paragraphs is similar

Lines 95-97: Another mini paragraph, please move it upwards and merge with the paragraphs from lines 76-85 at the end of it so that the paragraph structure would improve and it will have logical flow of presented idea.

Lines 86-94 and Lines 98-109: Paragraphs presented on these lines can be merged together, but would need to trim it further by providing relevant details only

MATERIALS AND METHODS: Material and method has been presented very well with elaborate details provided for each method analyzed during this experimental study. Experimental design was appropriately used and the statistical analysis was done very eloquently.

Line 117: “Grown” say “experimental” instead

Line 126: “established two” write “two established” instead

Line 129: “cold” do you mean controlled conditions? Please clarify and correct it accordingly. Also, year of evaluation is missing so please provide the same for better understanding

Lines 137-138: “for using them for glucosinolates analysis” rewrite it as “to use them for glucosinolates analysis”

Lines 126-138: Please avoid too many mini paragraphs, it does not add any value to the presented idea.

RESULTS: Authors have presented the result findings eloquently and all results are being presented in logical sequence. However, there are some concerns that must be addressed, please see the mentioned comments as shown below.

Lines 176-188: CL mentioned in the text in these lines are conflicting with CL defined in Table 1, please correct it accordingly.

Line 225: “and baby-leaves” there should be full stop at the end of the sentence, please correct it as needed.

Lines 227-228 and Lines 232-233: MGB in Lines 227-228 is defined as 4-methoxyglucobrassicin whereas in lines 232-232 it is defined as methoxyglucobrassicin. It looks like just another typo, please pay attention to such minor issues since they can likely change the context of result interpretation.

Line 234: “In the seeds the total GLSs”, there should be a comma after “In the seeds” and should be written as “In the seeds, the total GLSs….”

Lines 243-250: This paragraph is very confusing and inappropriate use of comma and poor sentence structure. Please rewrite this paragraph considering appropriate use of comma.

DISCUSSION: Authors have tried to discuss their findings and it is written well. There is unnecessary use of comma on many place throught the manuscript.

Line 282: “greenings”, please use some other alternative term for this

Line 282-287: This paragraph is poorly written and needs to revise to make the sentence structure more smooth and easily readable.

Line 293: “x 7 or x 2 times”, please rephrase them as “seven or two folds” instead

Lines 305-314: Please merge both paragraph’s or merge the mini paragraph of lines 305-307 with previous paragraph.

Lines 334-336: “The aim of this work has showed that the morphometric traits and the GLSs profile 334 were influenced by the climatic conditions that characterized the different plant growth 335 stages and by the genotype taken into consideration”. The conclusion is premature and without any strong evidence; authors must conduct this study on more than 3 environments for more than two years.

Line 347: “concentration” should be “concentrations”

Line 348: “that showed in this trial high level.” Later part of this sentence seems odd and awkward since what authors are trying to say doesn’t make sense. Please correct it as needed.

CONCLUSION: Authors have provided brief summary of their findings and key take home messages (THMs) have been provided. However, the first paragraph of conclusion seems overstated.

Line 359: “study” should be “studied”

Line 360: “in the study and confirming previous studies” should be written like “in this study” remove other part of the sentence since it makes the sentence very awkward

Line 369: “o” another typo, please correct it as needed.

TABLES AND FIGURES:

Table 1: The issue of CL must be addressed since it is highly confusing and readers are likely get distracted from the main findings.

Table 2: Same issues as reported in Table 1, please correct it accordingly.

Table 3: Line 219, in table caption SL is defined as Steam Length, this seems like a typo. Please correct it accordingly. 

Figure 1: Please increase the size of boxes shown on upper right corner of this figure, they are too small and very confusing.

Figure 2: Ditto, similar to Figure 1

Figure 3: Ditto, similar to Figure 1 and 2

Figure 4: Ditto, similar to Figure 1, 2, and 3

Note for Figure 1 and 2: Why different gradients have been used in both figures for same term such as NGB has two different gradients in both figures. At least I see them differently, please correct it whereever needed.

REFERENCES: Too many self citations, almost 26% references are self-citation (12 out of 49 references are self-citation- reference 4, 6, 7, 17, 26, 28, 31, 33, 34, 35, 37, and 45). Please remove unnecessary self-citations and only provide the relevant citations those are absolutely needed. If needed, please replace these self-citations with other relevant references.

Author Response

Comments and Suggestions for Authors

The manuscript is written well and it is of critical importance to stakeholders involved in this specialized crops of emerging market. However, there are some outstanding issues are noticeable and authors are encouraged to address them in the revised version.

Agronomy 1328925 Manuscript Review Report:

OVERALL REMARKS: The manuscript is written well and it is of critical importance to stakeholders involved in specialized crops of emerging market. However, there are some outstanding issues are noticeable and authors are encouraged to address them in the revised version. Please see the comments mentioned below for different sections.

A.A.: thank you for the positive comments and evaluation.

Comment 1. Additionally, similar abbreviation (CL) is being used for two different terms like “Cavolo Lacinato Nero di Toscana” and “Cotyledon Length”, please assign two different abbreviations respectively and correct it throught the manuscript.

A.A.: The correction was done. The “S” is used as an acronym for the cotyledon length.

Comment 2. Also, there is unnecessary use of comma on many place throught the manuscript. Please have your work reviewed from native English speaker to address this minor issues.

A.A.: The paper was revised by native English speaker. Attached to have the MPDI certificate.

ABSTRACT: Abstract is written succinctly and it is about in the recommended word count (196 words); however, there are abbreviations (CL and CR) being used without them spelled in the first place of occurance. The year and location of bio-agro-morphological evaluation needs to be mentioned briefly.

A.A.: The information were added.

INTRODUCTION: Authors have provided good background of related topics succinctly and cited relevant references. However, there are so many mini paragraphs, I encouraged authors to write one succinct medium length paragraphs that explains one idea rather than several mini paragraphs. This can be done to create three medium size paragraphs rather than having 8-10 mini-paragraphs. Also, please make sure to elaborate on research hypothesis in the last paragraph.

Line 54: “are increasing demanding options” please rewrite this as “are increasing in demand” instead

A.A.: The correction was done.

Lines 61-75: Please merge all three mini paragraphs into one since the context of these paragraphs is similar

A.A.: We modify the text according your suggestion.

Lines 76-85: Please merge both mini paragraphs into one since the context of these paragraphs is similar

A.A.: We modify the text according your suggestion.

Lines 95-97: Another mini paragraph, please move it upwards and merge with the paragraphs from lines 76-85 at the end of it so that the paragraph structure would improve and it will have logical flow of presented idea.

A.A.: We modify the text according your suggestion.

Lines 86-94 and Lines 98-109: Paragraphs presented on these lines can be merged together, but would need to trim it further by providing relevant details only

A.A.: We modify the text according your suggestion.

MATERIALS AND METHODS: Material and method has been presented very well with elaborate details provided for each method analyzed during this experimental study. Experimental design was appropriately used and the statistical analysis was done very eloquently.

Line 117: “Grown” say “experimental” instead

A.A: The correction was done

Line 126: “established two” write “two established” instead

A.A.: Done

Line 129: “cold” do you mean controlled conditions? Please clarify and correct it accordingly. Also, year of evaluation is missing so please provide the same for better understanding

A.A.: The information was added

Lines 137-138: “for using them for glucosinolates analysis” rewrite it as “to use them for glucosinolates analysis”

A.A: The correction was done

Lines 126-138: Please avoid too many mini paragraphs, it does not add any value to the presented idea.

A.A.: We modify the text according your suggestion.

RESULTS: Authors have presented the result findings eloquently and all results are being presented in logical sequence. However, there are some concerns that must be addressed, please see the mentioned comments as shown below.

Lines 176-188: CL mentioned in the text in these lines are conflicting with CL defined in Table 1, please correct it accordingly.

A.A: The correction was done

Line 225: “and baby-leaves” there should be full stop at the end of the sentence, please correct it as needed.

A.A.: Done

Lines 227-228 and Lines 232-233: MGB in Lines 227-228 is defined as 4-methoxyglucobrassicin whereas in lines 232-232 it is defined as methoxyglucobrassicin. It looks like just another typo, please pay attention to such minor issues since they can likely change the context of result interpretation.

A.A: The corrections were done

Line 234: “In the seeds the total GLSs”, there should be a comma after “In the seeds” and should be written as “In the seeds, the total GLSs….”

A.A.: Done

Lines 243-250: This paragraph is very confusing and inappropriate use of comma and poor sentence structure. Please rewrite this paragraph considering appropriate use of comma.

A.A.: The paragraph has been rewritten according your suggestion

DISCUSSION: Authors have tried to discuss their findings and it is written well. There is unnecessary use of comma on many place throught the manuscript.

A.A.: The discussion has been revised according your suggestion

Line 282: “greenings”, please use some other alternative term for this

A.A.: The sentence, according your suggestion was modified in “Various typologies of vegetables and herbs, including sprouts, microgreens, and ba-by-leaf, are becoming popular”

Line 282-287: This paragraph is poorly written and needs to revise to make the sentence structure more smooth and easily readable.

A.A.: The paragraph has been revised according your suggestion

Line 293: “x 7 or x 2 times”, please rephrase them as “seven or two folds” instead

A.A.: The correction was done

Lines 305-314: Please merge both paragraph’s or merge the mini paragraph of lines 305-307 with previous paragraph.

A.A.: The paragraph has been revised according your suggestion

Lines 334-336: “The aim of this work has showed that the morphometric traits and the GLSs profile were influenced by the climatic conditions that characterized the different plant growth stages and by the genotype taken into consideration”. The conclusion is premature and without any strong evidence; authors must conduct this study on more than 3 environments for more than two years.

A.A.: The discussion has been revised according your suggestion

Line 347: “concentration” should be “concentrations”

A.A.: Done

Line 348: “that showed in this trial high level.” Later part of this sentence seems odd and awkward since what authors are trying to say doesn’t make sense. Please correct it as needed.

A.A.: The sentence has been rewritten according your suggestion

CONCLUSION: Authors have provided brief summary of their findings and key take home messages (THMs) have been provided. However, the first paragraph of conclusion seems overstated.

A.A.: The discussion has been revised according your suggestion

Line 359: “study” should be “studied”

A.A.: Done

Line 360: “in the study and confirming previous studies” should be written like “in this study” remove other part of the sentence since it makes the sentence very awkward

A.A.: Done

Line 369: “o” another typo, please correct it as needed.

A.A.: Sorry for the mistake, the correction was done.

TABLES AND FIGURES:

Table 1: The issue of CL must be addressed since it is highly confusing and readers are likely get distracted from the main findings.

A.A.: The correction was done

Table 2: Same issues as reported in Table 1, please correct it accordingly.

A.A.: The correction was done

Table 3: Line 219, in table caption SL is defined as Steam Length, this seems like a typo. Please correct it accordingly.

A.A.: Sorry for the mistake, the correction was done.

Figure 1: Please increase the size of boxes shown on upper right corner of this figure, they are too small and very confusing.

A.A.: The size of boxes shown on upper right corner were increased.

Figure 2: Ditto, similar to Figure 1

A.A.: The size of boxes shown on upper right corner were increased.

Figure 3: Ditto, similar to Figure 1 and 2

A.A.: The size of boxes shown on upper right corner were increased.

Figure 4: Ditto, similar to Figure 1, 2, and 3

A.A.: The size of boxes shown on upper right corner were increased.

Note for Figure 1 and 2: Why different gradients have been used in both figures for same term such as NGB has two different gradients in both figures. At least I see them differently, please correct it whereever needed.

A.A.: The figure was modified.

REFERENCES: Too many self citations, almost 26% references are self-citation (12 out of 49 references are self-citation- reference 4, 6, 7, 17, 26, 28, 31, 33, 34, 35, 37, and 45). Please remove unnecessary self-citations and only provide the relevant citations those are absolutely needed. If needed, please replace these self-citations with other relevant references.

A.A.: The self citations were reduced.

Reviewer 2 Report

The manuscript entitled “Effects of the Growing Cycle and Genotype on the Morphometric and Glucosinolates Amount and Profile of Sprouts, Microgreens and Baby-leaves of Broccoli (Brassica oleracea L. var. Italica Plenck) and Kale (B. oleracea L. var. Acephala DC.) "Is an interesting and current one, but in order to be published, in my opinion, it needs some improvements"
Abstract: the purpose of this study is missing
2.2. Morphometric Parameters
The source Di Bella et al., 2020 is mentioned, but it is not found in the numbering of bibliographic references
2.4. Statistical Analysis
How many repetitions were used to analyze each sample?
Statistical analysis is incomplete in Tables 1, 2, 3
Discussions say that the results were influenced by climatic conditions. In this situation, the climatic data are missing.
The conclusion is general. It should be improved with specific conclusions from the manuscript

Author Response

Reviewer n. 2

The manuscript entitled “Effects of the Growing Cycle and Genotype on the Morphometric and Glucosinolates Amount and Profile of Sprouts, Microgreens and Baby-leaves of Broccoli (Brassica oleracea L. var. Italica Plenck) and Kale (B. oleracea L. var. Acephala DC.) "Is an interesting and current one, but in order to be published, in my opinion, it needs some improvements"

A.A.: Thank you for the positive comments and suggestions.

Abstract: the purpose of this study is missing

A.A.: The purpose of the study is added.

2.2. Morphometric Parameters

The source Di Bella et al., 2020 is mentioned, but it is not found in the numbering of bibliographic references

A.A.: Sorry for the mistake; the number (7) is added.

2.4. Statistical Analysis

How many repetitions were used to analyze each sample?

A.A.: Done

Statistical analysis is incomplete in Tables 1, 2, 3

A.A.: The information in tables 1,2, and 3 were added

Discussions say that the results were influenced by climatic conditions. In this situation, the climatic data are missing.

A.A.: The discussion was revised.

The conclusion is general. It should be improved with specific conclusions from the manuscript

A.A.: The conclusion was revised according to your suggestion

Round 2

Reviewer 1 Report

The authors have made significant revision and the manuscript looks much improved and can be considered to be accepted.

Reviewer 2 Report

Dear authors,

Manuscrisul "Effects of Growing Cycle and Genotype on the Morphometric Properties and Glucosinolates Amount and Profile of Sprouts, Microgreens and Baby Leaves of Broccoli (Brassica oleracea L. var. Italica Plenck) and Kale (B. oleracea L. var. Acephala DC.) "was revised according to my observations.

But in the future, make changes with track changes. It is very difficult to track each change if it is not visible.

Best regards!